# ActPerFL: Active Personalized Federated Learning

**Huili Chen, Jie Ding, Eric Tramel, Shuang Wu, Anit Kumar Sahu, Salman Avestimehr, Tao Zhang**

Alexa AI, Amazon

`{chehuili,jiedi,eritrame,wushuan,anitsah,avestime,taozhng}@amazon.com`

## Abstract

In the context of personalized federated learning (FL), the critical challenge is to balance local model improvement and global model tuning when the personal and global objectives may not be exactly aligned. Inspired by Bayesian hierarchical models, we develop Act-PerFL, a self-aware personalized FL method where each client can automatically balance the training of its local personal model and the global model that implicitly contributes to other clients' training. Such a balance is derived from the *inter-client and intra-client uncertainty quantification*. Consequently, Act-PerFL can adapt to the underlying clients' heterogeneity with *uncertainty-driven local training and model aggregation*. With experimental studies on Sent140 and Amazon Alexa audio data, we show that ActPerFL can achieve superior personalization performance compared with the existing counterparts.

## 1 Introduction

Federated learning (FL) (Konevcny et al., 2016; McMahan et al., 2017) is transforming machine learning (ML) ecosystems from "centralized in-the-cloud" to "distributed across-clients," to potentially leverage the computation and data resources of billions of edge devices (Lim et al., 2020), without raw data leaving the devices. As a distributed ML framework, FL aims to train a global model that aggregates gradients or model updates from the participating edge devices. Recent research in FL has significantly extended its original scope to address the emerging concern of personalization, a broad term that often refers to an FL system that accommodates client-specific data distributions of interest (Dinh et al., 2020a; Fallah et al., 2020a).

In particular, each client in a personalized FL system holds data that can be potentially non-IID. For example, smart edge devices at different houses may collect audio data of heterogeneous nature (Purington et al., 2017; Diao et al., 2020, 2021) due to, e.g., accents, background noises, and house structures. Each device hopes to improve its on-device model through personalized FL without transmitting sensitive data. While the practical benefits of personalization have been widely acknowledged, its theoretical understanding remains unclear. Existing works on personalized FL often derive algorithms based on a pre-specified optimization formulation or model aggregation rule.

In this work, we start with a toy example and develop insights into the nature of personalization from a statistical uncertainty perspective. In particular, we aim to answer the following critical questions regarding personalized FL.

*(Q1) The lower-bound baselines of personalized FL can be obtained in two cases, i.e., each client performs local training without FL, or all clients participate in conventional FL training. However, the upper-bound for the client is unclear.*

*(Q2) Suppose that the goal of each client is to improve its local model performance. How to design an FL training that interpret the global model, suitably aggregate local models and fine-tune each client's local training automatically?*

Both questions are challenging. The question (Q1) demands a systematic way to characterize the client-specific and globally-shared information. To this end, we draw insights from a simplified and analytically tractable setting: *two-level Bayesian hierarchical models*, where the two levels respectively describe inter-client and intra-client uncertainty.

We make the following technical contributions:

- Interpreting personalization from a *hierarchical model-based* perspective and providing theoretical analyses for FL training.

- Proposing ActPerFL, an active personalized FL solution that guides local training and global aggregation via inter- and intra-client *uncertainty quantification*.

- Presenting a novel implementation of Act-PerFL for deep learning, consisting of auto-

mated hyper-parameter tuning for clients and an adaptive aggregation rule.

- Evaluating ActPerFL on Sent140 and Amazon Alexa audio data. Empirical results show promising personalization performance compared with existing methods.

To our best knowledge, ActPerFL is the first work that utilizes uncertainty quantification to drive FL personalization.

## 2 Bayesian View of Personalized FL

We discuss how ActPerFL approaches personalized FL with theoretical insights from the Bayesian perspective in this section. To develop insights, we study a two-level Gaussian model. Similar arguments can be derived for generic parametric models. The notations are defined as follows. Let $\mathcal{N}(\mu, \sigma^2)$ denote Gaussian distribution with mean $\mu$ and variance $\sigma^2$. For a positive integer $M$, let $[M]$ denote the set $\{1, \ldots, M\}$. Let $\sum_{m \neq i}$ denote the summation over all $m \in [M]$ except for $m = i$. Suppose that there are $M$ clients.

From the server's perspective, it is postulated that data $z_1, \ldots, z_M$ are generated from the following two-layer Bayesian hierarchical model:

$$\theta_m \mid \theta_0 \stackrel{\text{IID}}{\sim} \mathcal{N}(\theta_0, \sigma_0^2), \quad z_m \mid \theta_m \stackrel{\text{IID}}{\sim} \mathcal{N}(\theta_m, \sigma_m^2),$$

for all clients with $m = 1, \ldots, M$. Here, $\sigma_0^2$ is a constant, and $\theta_0 \sim \pi_0(\cdot)$ is a hyperparameter with a non-informative flat prior. The above model represents both the connections and heterogeneity across clients. In particular, each client's data are distributed according to a client-specific parameter ($\theta_m$), which follows a distribution decided by a parent parameter ($\theta_0$). The parent parameter is interpreted as the root of shared information. Without loss of generality, we study client 1's local model as parameterized by $\theta_1$. Under the above model assumption, the parent parameter $\theta_0$ that represents the global model has a posterior distribution $p(\theta_0 \mid z_{1:M}) \sim \mathcal{N}(\theta^{(\text{G})}, v^{(\text{G})})$, where:

$$\theta^{(\text{G})} \triangleq \frac{\sum_{m \in [M]} (\sigma_0^2 + \sigma_m^2)^{-1} \theta_m^{(\text{L})}}{\sum_{m \in [M]} (\sigma_0^2 + \sigma_m^2)^{-1}}, \quad (1)$$

$$v^{(\text{G})} \triangleq \frac{1}{\sum_{m \in [M]} (\sigma_0^2 + \sigma_m^2)^{-1}}.$$

From the perspective of client $m$, we suppose that the postulated model is the same as above for $m = 2, \ldots, M$, and $\theta_1 = \theta_0$. It can be verified that the posterior distributions of $\theta_1$ without and with global Bayesian learning are $p(\theta_1 \mid z_1) \sim \mathcal{N}(\theta_1^{(\text{L})}, v_1^{(\text{L})})$ and $p(\theta_1 \mid z_{1:M}) \sim \mathcal{N}(\theta_1^{(\text{FL})}, v_1^{(\text{FL})})$, respectively, which can be computed as:

$$\theta_1^{(\text{L})} \triangleq z_1, \quad v_1^{(\text{L})} \triangleq \sigma_1^2,$$

$$\theta_1^{(\text{FL})} \triangleq \frac{\sigma_1^{-2} \theta_1^{(\text{L})} + \sum_{m \neq 1} (\sigma_0^2 + \sigma_m^2)^{-1} \theta_m^{(\text{L})}}{\sigma_1^{-2} + \sum_{m \neq 1} (\sigma_0^2 + \sigma_m^2)^{-1}}, \quad (2)$$

$$v_1^{(\text{FL})} \triangleq \frac{1}{\sigma_1^{-2} + \sum_{m \neq 1} (\sigma_0^2 + \sigma_m^2)^{-1}}.$$

The first distribution above describes the learned result of client 1 from its local data, while the second one represents the knowledge from all the clients' data in hindsight. Using the mean square error as risk, the Bayes estimate of $\theta_1$ or $\theta_0$ is the mean of the posterior distribution, namely $\theta_1^{(\text{L})}$ and $\theta_1^{(\text{FL})}$.

The flat prior on $\theta_0$ can be replaced with any other distribution to bake prior knowledge into the calculation. We consider the flat prior because the knowledge of the shared model is often vague in practice. The above posterior mean $\theta_1^{(\text{FL})}$ can be regarded as the optimal point estimation of $\theta_1$ given all the clients' data, thus is referred to as "FL-optimal". $\theta^{(\text{G})}$ can be regarded as the "global-optimal." The posterior variance quantifies the reduced uncertainty conditional on other clients' data. Specifically, we define the following *Personalized FL gain* for client 1 as:

$$\text{GAIN}_1 \triangleq \frac{v_1^{(\text{L})}}{v_1^{(\text{FL})}} = 1 + \sigma_1^2 \sum_{m \neq 1} (\sigma_0^2 + \sigma_m^2)^{-1}.$$

**Remark 1 (Posterior quantity interpretations)**
*Each client, say client 1, aims to learn $\theta_1$ in the personalized FL context. Its learned information regarding $\theta_1$ is represented by the Bayesian posterior of $\theta_1$ conditional on either its local data $z_1$ (without communications with others), or the data $z_{1:M}$ in hindsight (with communications). For the former case, the posterior uncertainty described by $v_1^{(\text{L})}$ depends only on the local data quality $\sigma_1^2$. For the latter case, the posterior mean $\theta_1^{(\text{FL})}$ is a weighted sum of clients' local posterior means, and the uncertainty will be reduced by a factor of $\text{GAIN}_1$. Since a point estimation of $\theta_1$ is of particular interest in practical implementations, we treat $\theta_1^{(\text{FL})}$ as the theoretical limit in the FL context (recall question Q1).*

**Remark 2 (Local training steps to achieve $\theta_1^{(\text{FL})}$)**
*Suppose that client 1 performs $\ell$ training steps using its local data and negative log-likelihood loss. We show that with a suitable number of steps and initial value, client 1 can obtain the intended $\theta_1^{(\text{FL})}$. The local objective is:*

$$\theta \mapsto (\theta - z_1)^2 / (2\sigma_1^2) = (\theta - \theta_1^{(\text{L})})^2 / (2\sigma_1^2), \quad (3)$$

*which coincides with the quadratic loss. Let $\eta \in (0, 1)$ denote the learning rate. By running the gradient descent:*

$$\theta_1^\ell \leftarrow \theta_1^{\ell-1} - \eta \frac{\partial}{\partial \theta} \left( (\theta - \theta_1^{(\text{L})})^2 / (2\sigma_1^2) \right) |_{\theta_1^{\ell-1}}$$

$$= \theta_1^{\ell-1} - \eta (\theta_1^{\ell-1} - \theta_1^{(\text{L})}) / \sigma_1^2 \quad (4)$$

*for $\ell$ steps with initial value $\theta_1^{\text{INIT}}$, client 1 obtains:*

$$\theta_1^\ell = \left(1 - (1 - \sigma_1^{-2}\eta)^\ell\right)\theta_1^{(\text{L})} + (1 - \sigma_1^{-2}\eta)^\ell\theta_1^{\text{INIT}}. \quad (5)$$

*It can be verified that Eqn. (5) becomes $\theta_1^{(\text{FL})}$ in Eqn. (2) if and only if:*

$$\theta_1^{\text{INIT}} = \frac{\sum_{m\neq 1}(\sigma_0^2 + \sigma_m^2)^{-1}\theta_m^{(\text{L})}}{\sum_{m\neq 1}(\sigma_0^2 + \sigma_m^2)^{-1}}, \quad (6)$$

$$(1 - \sigma_1^{-2}\eta)^\ell = \frac{\sum_{m\neq 1}(\sigma_0^2 + \sigma_m^2)^{-1}}{\sigma_1^{-2} + \sum_{m\neq 1}(\sigma_0^2 + \sigma_m^2)^{-1}}. \quad (7)$$

*In other words, with a suitably chosen initial value $\theta_1^{\text{INIT}}$, learning rate $\eta$, and the number of (early-stop) steps $\ell$, client 1 can obtain the desired $\theta_1^{(\text{FL})}$.*

## 3 Proposed Solution for Personalized FL

Our proposed ActPerFL framework has three key components as detailed in this section: (i) proper initialization for local clients at each round, (ii) automatic determination of the local training steps, (iii) discrepancy-aware aggregation rule for the global model. These components are interconnected and contribute together to ActPerFL's effectiveness. Note that points (i) and (iii) direct ActPerFL to the regions that benefit personalization in the optimization space during local training, which is not considered in prior works such as DITTO (Li et al., 2021) and pFedMe (Dinh et al., 2020b). Therefore, ActPerFL is more than imposing implicit regularization via early stopping.

In this section, we show how the posterior quantities of interest in Section 2 can be connected with FL. Recall that each client $m$ can obtain the FL-optimal solution $\theta_m^{(\text{FL})}$ with the initial value $\theta_m^{\text{INIT}}$ in Eqn. (6) and tuning parameters $\eta, \ell$ in Eqn. (7). Also, it can be shown that $\theta_m^{\text{INIT}}$ is connected with the global-optimal $\theta^{(\text{G})}$ in Eqn. (1) through

$$\theta_m^{\text{INIT}} = \theta^{(\text{G})} - \frac{(\sigma_0^2 + \sigma_m^2)^{-1}}{\sum_{k:\,k\neq m}(\sigma_0^2 + \sigma_k^2)^{-1}}(\theta_m^{(\text{L})} - \theta^{(\text{G})}). \quad (8)$$

The initial value $\theta_m^{\text{INIT}}$ in Eqn. (8) is unknown during training since $\theta_m^{(\text{L})}, \theta^{(\text{G})}$ are both unknown. A natural solution is to update $\theta_m^{\text{INIT}}$, $\theta_m^{(\text{L})}$, and $\theta^{(\text{G})}$ iteratively, leading to the following personalized FL rule of our ActPerFL framework.

**Generic ActPerFL.** At the $t$-th ($t \geq 1$) round:

• *Client $m$* receives the latest global model $\theta^{t-1}$ from the server (initialized as $\theta^0$), and calculates:

$$\theta_m^{t,\text{INIT}} \triangleq \theta^{t-1} - \frac{(\sigma_0^2 + \sigma_m^2)^{-1}(\theta_m^{t-1} - \theta^{t-1})}{\sum_{k:\,k\neq m}(\sigma_0^2 + \sigma_k^2)^{-1}}, \quad (9)$$

where $\theta_m^{t-1}$ is client $m$'s latest personal parameter at round $t-1$, initialized to be $\theta^0$. Starting

from the above $\theta_m^{t,\text{INIT}}$, client $m$ performs gradient descent-based local updates with optimization parameters following Eqn. (7) or its approximations, and obtains a personal parameter $\theta_m^t$.

• *Server* collects $\theta_m^t$ and calculates:

$$\theta^t \triangleq \frac{\sum_{m\in[M]}(\sigma_0^2 + \sigma_m^2)^{-1}\theta_m^t}{\sum_{m\in[M]}(\sigma_0^2 + \sigma_m^2)^{-1}}. \quad (10)$$

In general, the above $\sigma_0^2, \sigma_m^2$ represent "inter-client uncertainty" and "intra-client uncertainty," respectively. When $\sigma_0^2$ and $\sigma_m^2$'s are unknown, they can be approximated asymptotically or using practical finite-sample approximations.

**SGD-based practical algorithm for DL.** For the above training method, the quantities $\sigma_0^2$ and $\sigma_m^2$ are crucial as they affect the choice of learning rate $\eta_m$ and the early-stop rule. However, these two values are unknown in complex learning models. To approximate the uncertainty quantities, we generally treat $\sigma_m^2$ as "uncertainty of the local optimal solution $\theta_m^{(\text{L})}$ of client $m$", and $\sigma_0^2$ as "uncertainty of clients' underlying parameters." Assume that for each client $m$, we had $u$ independent samples of its data and the corresponding local optimal parameter $\theta_{m,1}, \ldots, \theta_{m,u}$. We could then estimate $\sigma_m^2$ by their *sample variance*. In particular, at round $t$, we approximate $\sigma_m^2$ with:

$$\widehat{\sigma_m^2} = \text{empirical variance of } \{\theta_m^1, \ldots, \theta_m^t\}. \quad (11)$$

Likewise, at round $t$, we estimate $\sigma_0^2$ by:

$$\widehat{\sigma_0^2} = \text{empirical variance of } \{\theta_1^t, \ldots, \theta_M^t\}. \quad (12)$$

For multi-dimensional parameters, we introduce the following counterpart uncertainty measures. For vectors $x_1, \ldots, x_M$, their empirical variance is defined as the trace of $\sum_{m\in[M]}(x_m - \bar{x})(x_m - \bar{x})^{\text{T}}$, which is the sum of entry-wise empirical variances. $\widehat{\sigma_m^2}$ and $\widehat{\sigma_0^2}$ are defined from such empirical variances similar to Eqn. (11) and (12). The above quantities can be calculated recursively online with constant memory (Han et al., 2017). Alg. 1 outlines the workflow of ActPerFL.

## 4 Experimental Studies

**Experimental setup.** We evaluate ActPerFL's performance on two NLP datasets: Sentiment140 (Go et al., 2009) and private Amazon Alexa audio data. Sent140 is a text sentiment analysis dataset with two output classes and 772 clients. We generate non-i.i.d. data following FedProx (Li, 2020). The audio dataset is collected for wake-word detection task (i.e., binary classification). This dataset contains 39 thousand hours of training data and 14 thousand hours of test data. We use a two-layer

**Algorithm 1** Active Personal FL (ActPerFL)

**Input:** A server and $M$ clients. Communication rounds $T$, client activity rate $C$, client $m$'s local data $D_m$ and learning rate $\eta_m$.

**for** each communication round $t = 1, \ldots T$ **do**
    Sample clients: $\mathbb{M}_t \leftarrow \max(\lfloor C \cdot M \rfloor, 1)$
    **for** each client $m \in \mathbb{M}_t$ **in parallel do**
        Distribute server model $\theta^{t-1}$ to client $m$
        Estimate $\widehat{\sigma_m^2}$ using Eqn. (11)
        Compute local step $l_m$ from Eqn. (7) and local initialization $\theta_m^{\text{INIT}}$ via Eqn. (6)
        $\theta_m^t \leftarrow LocalTrain(\theta_m^{\text{INIT}}, \eta_m, l_m; D_m)$
    Server estimates $\widehat{\sigma_0^2}$ using Eqn. (12)
    Server updates global model $\theta^t$ via Eqn. (10)

LSTM model and an 11-layer CNN model for these two datasets, respectively. For comparison, we also evaluate the personalization performance of FedAvg (McMahan et al., 2017), DITTO (Li et al., 2021), PerFedAvg (Fallah et al., 2020b), and pFedMe (Dinh et al., 2020b).

### 4.1 Results on Alexa Audio Data

For Alexa audio data, we use a CNN that is pre-trained on the training data of different device types (i.e., heterogeneous data) as the initial global model to *warm-start* FL training. The personalization task aims to improve the wake-word detection performance at the *device type level*. We assume there are five clients in the FL system and all of them participate in each round. Each client has the training data for a specific device type.

**Evaluation metric.** We evaluate the performance using the pre-trained model (for warm-start) as the *baseline*. To compare different FL algorithms, we use the *relative false accept (FA)* value of the resulting model when the associated relative false reject (FR) is close to one as the metric. So a smaller relative FA is preferred. Here, the relative FA and FR are computed using the baseline.

For comparison, we implement FedAvg and DITTO with both equal-weighted and sample size-based model averaging (denoted by the suffix '-e' and '-w', respectively) during aggregation. For PerFedAvg (Fallah et al., 2020b), we use its first-order approximation and the equal-weighted aggregation. We did not report pFedMe (Dinh et al., 2020b) due to its divergence with various hyper-parameters. Table 1 summarizes the performance of the updated global model. The results show that ActPerFL achieves the lowest relative FA, thus obtaining the best global model. We further compare

Table 1: Detection performance (relative FA) of the *global model* on the test dataset.

| FL methods | Device Types | | | | |
|---|---|---|---|---|---|
| | A | B | C | D | E |
| **ActPerFL** | **0.92** | **0.94** | **0.91** | **0.91** | 1.01 |
| **FedAvg-w** | 8.39 | 4.00 | 12.80 | 8.61 | 10.62 |
| **FedAvg-e** | 0.97 | 0.96 | 1.00 | 0.92 | 1.00 |
| **DITTO-w** | 8.38 | 4.00 | 12.75 | 8.61 | 10.23 |
| **DITTO-e** | 0.97 | 0.95 | 1.00 | 0.93 | **0.99** |
| **PerFedAvg** | 1.06 | 0.98 | 1.08 | 0.93 | 1.01 |

Table 2: Detection performance (relative FA) of the *personalized models* on a test dataset.

| FL methods | Device Type | | | | |
|---|---|---|---|---|---|
| | A | B | C | D | E |
| **ActPerFL** | **0.93** | **0.91** | **0.90** | **0.90** | 0.99 |
| **FedAvg-e** | 0.95 | 0.95 | 0.93 | 0.91 | 0.98 |
| **DITTO-e** | 0.97 | 0.96 | 0.93 | 0.91 | 0.96 |
| **PerFedAvg** | 1.02 | 1.11 | 1.08 | 1.00 | **0.93** |

the personalization performance of local models obtained by different FL algorithms in Table 2.

### 4.2 Results on Sent140 Text Data

In this experiment, we also use warm-start by training a global model from scratch with FedAvg for 200 rounds for initializing other FL algorithms. Then, we continue FL training with various FL methods for another 400 rounds. Figure 1 compares the training and test accuracy of the personalized models obtained by different FL algorithms where the accuracy is aggregated across clients. We can see that both ActPerFL and FedAvg demonstrate better convergence performance compared to DITTO (Li et al., 2021), pFedMe (Dinh et al., 2020b), and PerFedAvg (Fallah et al., 2020b).

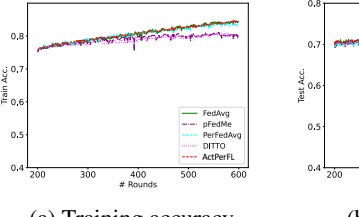 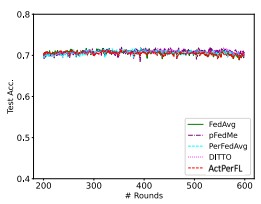

(a) Training accuracy.      (b) Test accuracy.

Figure 1: Performance of FL methods on Sent140 data.

### 5 Concluding Remarks

We proposed ActPerFL to address the challenge of balancing local model training and global model aggregation in personalized FL. Our solution adaptively adjusts local training with automated hyper-parameter selection and performs uncertainty-weighted global aggregation. Empirical studies show that ActPerFL can achieve promising performance on NLP applications.

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
