# OpenReview forum: "ActPerFL: Active Personalized Federated Learning"
_aclweb.org/ACL/2022/Workshop/FL4NLP — FL4NLP@ACL2022_

### Official Review · Reviewer_S4x7 · 2022-03-22
**Interesting perspective on personalized FL**

**Rating:** 7
**Confidence:** 4

**Review:**

Summary: The work presents a framework to analyze personalized FL with bayesian models where the posterior mean for each FL (collaborative training) and local training are theoretically analyzed so that the uncertainty of each model can be quantified, and utilized for maximum personalized FL performance. With this framework, the paper gives theoretical insights into how PFL performance can be maximized by the proper initialization, number of local steps, and global model aggregation. Empirical experiments are done on Alexa Audio Data and Sent140 Data for validation of the proposed algorithm.

Pros:
- The paper looks at an important problem of personalization in FL in the views of Bayesian models which has not been looked into a lot before.
- The paper is clear to read including the mathematical notations and description of the algorithms.
- The paper gives a concrete analysis into what kind of parameters (initialization, local steps, and aggregation) affect the personalized FL performance in a Bayesian framework and the exact forms of those parameters to maximize personalized FL performance.
- The paper includes empirical validation on two different datasets.

Cons:
- Although the theoretical insights are interesting, they are very difficult to implement in practice due to the pre-knowledge on the variance (uncertainty) for deciding the initialization and local steps. However, despite this difficulty, the AdaPerFL that the paper proposes is able to use the sample-variance to circumvent this problem and achieve reasonable performance, thus managing to compensate for this drawback.

---

### Official Review · Reviewer_8xw5 · 2022-03-24
**Interesting Bayesian formulation of the personalized federated learning problem.**

**Rating:** 5
**Confidence:** 3

**Review:**

Summary:
In this work, the authors introduce a Bayesian formulation of the personalized federated learning problem and provide theoretical insights on how to quantify (inter- and intra-) clients uncertainty during federated training. Their propose solution, AdaPerFL, views the personalization problem from the lens of a two-layer Bayesian hierarchical model that can lead to an automatic determination of the local training steps of each client and a heterogeneous- and uncertainty-aware global model aggregation rule.

Strong and Weak Points:

(S1) Interesting Bayesian formulation of personalized FL.
(S2) Evaluation over a range of existing baselines.

(W1) Related work and additional background/preliminary information is missing.
(W2) Provided evaluation does not show AdaPerFL's improved performance.
(W3) Missing details on experimental evaluation.


(W1) A section that provides more background knowledge in the context of personalized FL (e.g., [1])
and discussion on existing Bayesian Federated Learning formulations (e.g., [2]) is missing.
Section 2 could be summarized and placed in an Appendix in favor of such a section. Within the
proposed section it would also be better to discuss how your approach is compared to the experimental
baselines so that non-expert readers can understand the trade-offs of the different approaches. Moreover,
in equation (1) it is not clear how the definitions are derived (sum over posterior distributions?). Also,
how is the local objective equation (3) derived? Even though the authors discuss uncertainty quantification
through GAIN, the metric is never used in the rest of the paper. What is the update rule of LocalTrain
in Algorithm 1? Is it the local update rule presented in equation (4)?

(W2) Equally weighted FedAvg seems to provide the same performance as AdaPerFL in the wake-word detection task.
Similar for the DITTO-e. Can the authors elaborate more on these results? Moreover, there do not seem to be any
improvements on the sentiment analysis task both in terms of testing and training performance. Maybe, it would
have been better to demonstrate the results as in the case of the Alexa Audio domain by randomly sampling a set
of clients from the available pool of 772 clients and showing the performance of the global and personalized
models on a test dataset. Additionally, it is not clear why pretraining was required for both learning tasks
(first task is reasonable since it is harder domain bit why for the second?), maybe that reasons the learning
behavior in the sentiment analysis task (i.e., similar performance for all approaches)? Table 1 and Figure 1
have Self-FL as a method which is never introduced; I suspect that this the AdaPerFL.

(W3) For the first task you only consider 5 clients for a federation. Why a so small number?  What is the distribution
of training/valid/test sets at each client (s/th like {'train': #, 'valid': #, 'test': #} would be appropriate) in this task?
Moreover, your proposed method is an adaptive learning approach, what is the distribution of local steps per client in your
experiments? Is there a particular range of local step values that each client or all clients follow? To assign the local steps
to each client do you use the empirical estimations of equations (11) and (12) in equation (7)? What is the learning rate
you used for training?

[1] Wu, Jinze, Qi Liu, Zhenya Huang, Yuting Ning, Hao Wang, Enhong Chen, Jinfeng Yi, and Bowen Zhou.
"Hierarchical personalized federated learning for user modeling." In Proceedings of the Web Conference 2021, pp. 957-968. 2021.

[2] Yurochkin, Mikhail, Mayank Agarwal, Soumya Ghosh, Kristjan Greenewald, Nghia Hoang, and Yasaman Khazaeni.
"Bayesian nonparametric federated learning of neural networks." In International Conference on Machine Learning, pp. 7252-7261. PMLR, 2019.

---

### Official Review · Reviewer_RVNp · 2022-03-26
**Novel and interesting approach to personalization. Extension from mean estimation to non-convex loss is not clear.**

**Rating:** 7
**Confidence:** 3

**Review:**

Originality: High.
AdaPerFL uses a novel, interesting and rigorous Bayesian approach to evaluate Personalization in Federated Learning (FL). To motivate the framework, the authors assume a generative hierarchical Gaussian model. FL is used for mean estimation. The authors quantify the benefit of FL in this setting (compared to a local estimation) as Personalized FL gain.
The concepts of inter-client certainty and intra-client certainty are very natural and potentially useful.
Equation (10) could be a very interesting approach to global model aggregation and feels natural (inversely weighting by variance of each local estimate)

Clarity: Low.
The first part of the paper is very clear - generative Gaussian model.
Everything after Remark 2 is not very clear. And why should optimal initialization, learning rate and #steps that are useful for a specific convex problem (mean estimation in a hierarchical Gaussian) be applicable to non-convex deep learning? Why is equation (10) useful? How does convergence to a stationary point proceed if we use eq (10)?

The empirical evaluation is also not clear. On Sent140, the authors claim that AdaPerFL is better than Ditto, but the graphs shows test accuracy to be the same. A table here would make things much clearer. On Alexa, the main benefit seems to be using equal weighted averaging instead of example weighted averaging. The benefits between FedAvg-e and AdaPerFL is small, but the difference between FedAvg-e and FedAvg-w is very large (not observed by other papers).

Significance: Medium.
This is a very interesting direction and approach to FL and Personalization. However, the extension to non-convex problems is not clear. And the experimental evaluation can be improved.

Quality: Medium.
Overall, I'd say the quality is medium. There is an original and potentially significant contribution here. The authors could improve the quality by proving convergence behavior of equation (10), and showing why the choice of {LR, initialization, step size} for mean estimation are useful in completely different problems that are not even convex. The experimental evaluation can also be much better.

---

### Decision · Program_Chairs · 2022-03-26

Accept